# Pre-admission interventions (prehabilitation) to improve outcome after major elective surgery: a systematic review and meta-analysis

Rachel Perry ![ORCID],[1] Georgia Herbert,[1] Charlotte Atkinson,[1] Clare England,[1,2] Kate Northstone,[3] Sarah Baos,[4] Tim Brush,[4] Amanda Chong,[1] Andy Ness,[1,5] Jessica Harris,[4] Anne Haase,[6] Sanjoy Shah,[7] Maria Pufulete[4]

For numbered affiliations see end of article.

**Correspondence to**
Dr Maria Pufulete;
maria.pufulete@bristol.ac.uk

## ABSTRACT

**Objective** To determine the benefits and harms of pre-admission interventions (prehabilitation) on postoperative outcomes in patients undergoing major elective surgery.

**Design** Systematic review and meta-analysis of randomised controlled trials (RCTs) (published or unpublished). We searched Medline, Embase, CENTRAL, DARE, HTA and NHS EED, The Cochrane Library, CINAHL, PsychINFO and ISI Web of Science (June 2020).

**Setting** Secondary care.

**Participants** Patients (≥18 years) undergoing major elective surgery (curative or palliative).

**Interventions** Any intervention administered in the preoperative period with the aim of improving postoperative outcomes.

**Outcomes and measures** Primary outcomes were 30-day mortality, hospital length of stay (LoS) and postoperative complications. Secondary outcomes included LoS in intensive care unit or high dependency unit, perioperative morbidity, hospital readmission, postoperative pain, heath-related quality of life, outcomes specific to the intervention, intervention-specific adverse events and resource use.

**Review methods** Two authors independently extracted data from eligible RCTs and assessed risk of bias and the certainty of evidence using Grading of Recommendations, Assessment, Development and Evaluation. Random-effects meta-analyses were used to pool data across trials.

**Results** 178 RCTs including eight types of intervention were included. Inspiratory muscle training (IMT), immunonutrition and multimodal interventions reduced hospital LoS (mean difference vs usual care: −1.81 days, 95% CI −2.31 to −1.31; −2.11 days, 95% CI −3.07 to −1.15; −1.67 days, 95% CI −2.31 to −1.03, respectively). Immunonutrition reduced infective complications (risk ratio (RR) 0.64 95% CI 0.40 to 1.01) and IMT, and exercise reduced postoperative pulmonary complications (RR 0.55, 95% CI 0.38 to 0.80, and RR 0.54, 95% CI 0.39 to 0.75, respectively). Smoking cessation interventions reduced wound infections (RR 0.28, 95% CI 0.12 to 0.64).

**Conclusions** Some prehabilitation interventions may reduce postoperative LoS and complications but the quality of the evidence was low.

**PROSPERO registration number** CRD42015019191.

## Strengths and limitations of this study

► Unlike previous systematic reviews that focused on single interventions in single surgical populations, this review provides a summary of all types of prehabilitation interventions across *all* surgical populations.

► Comprehensive methods, with inclusion of published literature in all languages, alongside grey literature searching, to avoid publication bias.

► The large number of meta-analyses performed for related outcomes with data from the same individuals may lead to effect size multiplicity.

## INTRODUCTION

There are over 1 500 000 major surgical procedures carried out in the UK each year, with an annual cost of about £5.6 billion.[1] An increasing proportion of surgical patients are high risk, as they are elderly, frail, obese and have multiple comorbidities. Modifiable factors increase the risk of death and complications after surgery (which affect up to 75% of patients[2] and reduce quality of life (QoL)).[3–5] The implementation of enhanced recovery after surgery (ERAS), also known as 'fast-track surgery', has led to considerable improvements in patient care,[6] although these programmes have largely focused on optimising the surgical and recovery pathways in hospital with little focus on preoperative patient optimisation.

Prehabilitation is a broad term applied to interventions administered prior to surgery to improve health and fitness with the aim of reducing surgery-related morbidity and facilitating recovery. Prehabilitation programmes include physical activity,[7–13] nutrition support,[14] smoking cessation,[15] alcohol cessation,[16] respiratory interventions (eg, incentive spirometry (IS) and inspiratory muscle

training (IMT)),[17] education[18] and combined interventions.[19] The inclusion of prehabilitation to standard ERAS programmes could allow patients to optimise their eligibility for surgery and further improve their outcomes.[20]

There are a large number of systematic reviews of prehabilitation, but most of these have focused on a single intervention for a specific surgical group (eg, exercise in cancer surgery[21]; IMT in cardiac surgery[22]; immunonutrition in head and neck surgery[23]; etc). This is despite the fact that risk factors for surgical complications are similar across all types of major surgery and, generally, clinicians in preoperative assessment clinics (where prehabilitation is likely to be implemented) assess and treat all patients, regardless of type of surgery patients undergo. The objectives of this systematic review were to (a) identify all interventions that have been administered prior to any major elective surgery, (b) evaluate the potential benefits and harms of these interventions, and (c) compare the effectiveness of the different interventions on postoperative outcomes.

## METHODS

The protocol for this review was published previously.[24] We included all published and unpublished randomised controlled trials (RCTs) with the following characteristics: (1) *Participants*: adult patients (≥18 years) undergoing major elective surgery (under general anaesthesia resulting in a minimum hospital stay of at least 2 days), excluding day case surgery; (2) *Interventions*: administered before elective surgery with the aim of improving short-term (up to 3 months) postoperative outcomes. Interventions administered for less than 24 hours before surgery and/or continued postoperatively, studies focusing on ERAS, studies of enteral (via nasogastric tube) or parenteral nutrition, and intravenous drug administration were excluded. Studies in which the intervention was designed to improve a functional outcome specific to one type of surgery (eg, knee exercises to improve movement of the knee after arthroplasty) were excluded, as these were not deemed to represent a generic improvement of functional capacity. The comparator was no intervention or usual care; studies comparing different prehabilitation interventions or where the comparator deviated substantially from usual care were excluded. (3) *Outcomes*: primary outcomes included mortality (30 days), hospital length of stay (LoS) and postoperative complications (infective and non-infective). Secondary outcomes included LoS in intensive care unit or high dependency unit, perioperative morbidity, hospital readmission, postoperative pain, health-related quality of life (QoL), outcomes specific to the intervention, intervention-specific adverse events and resource use.

### Identification of studies

The following electronic databases were searched up to June 2020, with no language restrictions: Medline and PreMedline (OvidSP) (1950 to date), Embase Classic+Embase (OvidSP) (1974 to date), CENTRAL, DARE, HTA and NHS EED (The Cochrane Library, latest Issue), CINAHL (1981 to date), PsychINFO (1806 to date), ISI Web of Science: Science Citation Index Expanded (1900 to date), ISI Web of Science: Conference Proceedings Citation Index-Science (1990 to date), Current Controlled Trials (www.controlled-trials.com with links to other databases of ongoing trials) and the WHO International Clinical Trials Registry Platform (www.who.int/ictrp/en/).

Reference lists of included studies and reviews were hand searched. OpenGrey, Google (to page 10), non-indexed journals, theses and dissertations, and published protocols were also searched until June 2020. Experts in the field and trial authors were contacted for further information or unpublished data. The search strategy for Medline is shown in the online supplemental digital content; this was adapted as appropriate for searching other databases.

### Study selection and data extraction

One review author independently screened all titles and abstracts for eligibility (RP). A second review author screened a randomly selected sample (10%) (MP). Two review authors (RP and MP) independently assessed all full-text papers for eligibility. Any disagreements were resolved by discussion and consensus with a third review author (CA). Reasons for excluding studies were recorded (see online supplemental digital content). Relevant data were extracted by multiple independent reviewers (each study had two independent reviewers) and all data extractions were checked and moderated by RP, MP and CA. Where multiple papers reported the same study but different outcomes, all sources were used for data extraction.

### Risk of bias

Risk of bias (RoB) was independently assessed by two review authors using The Cochrane Collaboration RoB tool.[25] Outcomes were grouped into outcome domains and pragmatic a priori decisions were made when assessing RoB for each of these domains. For example, lack of blinding of outcome assessors was not deemed to be able to influence objective outcomes such as mortality, so this outcome was judged to be at low risk of detection bias irrespective of blinding (see online supplemental material for RoB tool adapted for this review). The strength of the overall body of evidence for each outcome was assessed using the Grading of Recommendations, Assessment, Development and Evaluation (GRADE) methodology.[26]

### Data synthesis

Results for trials that used variations of similar interventions (eg, different types of physical activity, psychological or educational programmes) were grouped. Due to heterogeneity in participants, interventions and intervention delivery, a random-effects meta-analysis was used for

the primary analysis when pooling data across trials. Meta-analyses were only conducted if data from three or more trials were available. Findings from studies that were not meta-analysed were summarised narratively. Fixed-effects meta-analysis was used as a secondary analysis. Reasons for missing data were recorded (eg, drop-outs, losses to follow-up and withdrawals).

Where primary outcome data were not provided in the form of a mean and SD, we derived these from the reported test statistics (eg, SD from SEs or 95% CIs) or estimated them (eg, mean and SD from median and range). We used the following methods to estimate or impute missing data: (1) where LoS aggregate data were presented as median and range, we estimated mean and SD using the formulae described by McGrath *et al*[27]; (2) we imputed missing SDs for LoS using the mean of the SDs reported by other studies within that treatment arm; (3) where LoS data were presented as Kaplan-Meier graphs, we extracted the following LoS data for each trial arm where available: median (50%), IQR (25%–75%) and range (minimum and maximum). Mean LoS and its associated SD were subsequently derived as described above. We did not use results for LoS presented as HRs without further descriptive LoS measures to estimate median LoS, due to potentially high uncertainty in estimation[28]; (4) where complications were reported as per cent incidence, we converted this into the number of participants who experienced complications.

Pooled risk ratios (RRs) and 95% CIs were calculated for dichotomous outcomes using the Mantel-Haenszel method for both random-effects and fixed-effects meta-analyses. Pooled mean differences and 95% CIs or standardised mean differences and 95% CI were calculated for continuous outcomes (LoS) using the inverse-variance method (for both random-effects and fixed-effects meta-analyses), when results were reported on the same scale (or could be converted to the same scale) or if results were reported on different scales, respectively. The unit of analysis in all included studies was the individual participant. No studies used cluster randomisation. Funnel plots were used to assess publication bias when 10 or more studies had been included in a meta-analysis. We formally tested for funnel plot asymmetry using the Egger's regression test. All plots are shown in the online supplemental material.

## Assessment of heterogeneity
Clinical heterogeneity across studies was assessed by examining variability in participants, baseline data, interventions and outcomes. Statistical heterogeneity was quantified using the $I^2$ statistic. We applied the following thresholds for the interpretation of the $I^2$ statistic[29]: 0%–40% might not be important; 30%–60% may represent moderate heterogeneity; 50%–90% may represent substantial heterogeneity; 75%–100% represents considerable heterogeneity.

## Sensitivity analyses
We prespecified three sensitivity analyses: (1) including only trials classified as 'low risk' for random sequence generation and allocation concealment; (2) excluding studies with imputed results; (3) fixed-effects meta-analyses.

## Subgroup analyses
We prespecified the following subgroup analyses: (1) type of surgery (eg, orthopaedic, cardiac, abdominal); (2) cancer versus non-cancer surgery; (3) type of intervention (eg, brief, 5 days or less vs longer term, more than 5 days); (4) intervention conducted pre-ERAS or post-ERAS implementation; (5) high-risk versus low-risk surgical patients. All analyses were performed on RevMan (V.5.3).

## Patient and public involvement
No patients were involved in the design, conduct, and analysis of the study and interpretation of findings, although prehabilitation was identified as one of the priority themes for research by the James Lind Alliance Heart Surgery Priority Setting Partnership and the James Lind Alliance Anaesthesia and Perioperative Care Priority Setting Partnership, which included patient stakeholder groups.

## RESULTS
One hundred ninety-four articles pertaining to 178 studies were eligible for inclusion (figure 1). Of these, 29 were unpublished studies (and 5 of these were included in meta-analyses). A summary of the interventions and main results is shown in table 1. The characteristics of the included studies by type of intervention, aggregate data from the included studies by type of intervention, RoB assessment, summary of findings, excluded studies, studies that fit the inclusion criteria but had no usable data, and relevant protocols of ongoing or unpublished studies are shown in the online supplemental digital content. We identified eight types of interventions administered in the preoperative period: nutritional (51 studies),[30–80] respiratory (30 interventions of 29 studies)[81–109] exercise (27 studies),[110–136] multimodal (25 studies),[100 108 137–159] educational (17 studies),[160–176] psychological (16 studies),[177–192] smoking and alcohol cessation (7 studies)[193–199] and pharmacological (5 studies)[200–204] (table 1).

Nutritional interventions were further subdivided into standard oral nutritional supplements (ONS, 7 studies),[30–35 69] oral immunonutrition supplements (19 studies)[36–47 70–75 80] weight loss interventions (11 studies),[48–51 53–57 76 205] oral prebiotics and probiotics (6 studies),[58–61 77 78] dietary optimisation of comorbidities (3 studies)[62–64] and other (5 studies, generally including administration of nutritional supplements such as fish oils, antioxidants, etc).[65–68 79] Respiratory interventions were further subdivided into IMT (18 studies),[81–95 106 108 109] IS (5 studies)[90 96–99] and combined respiratory interventions (7 studies, involving combinations of IMT, IS, other respiratory exercises and physical exercises).[100–105 107]

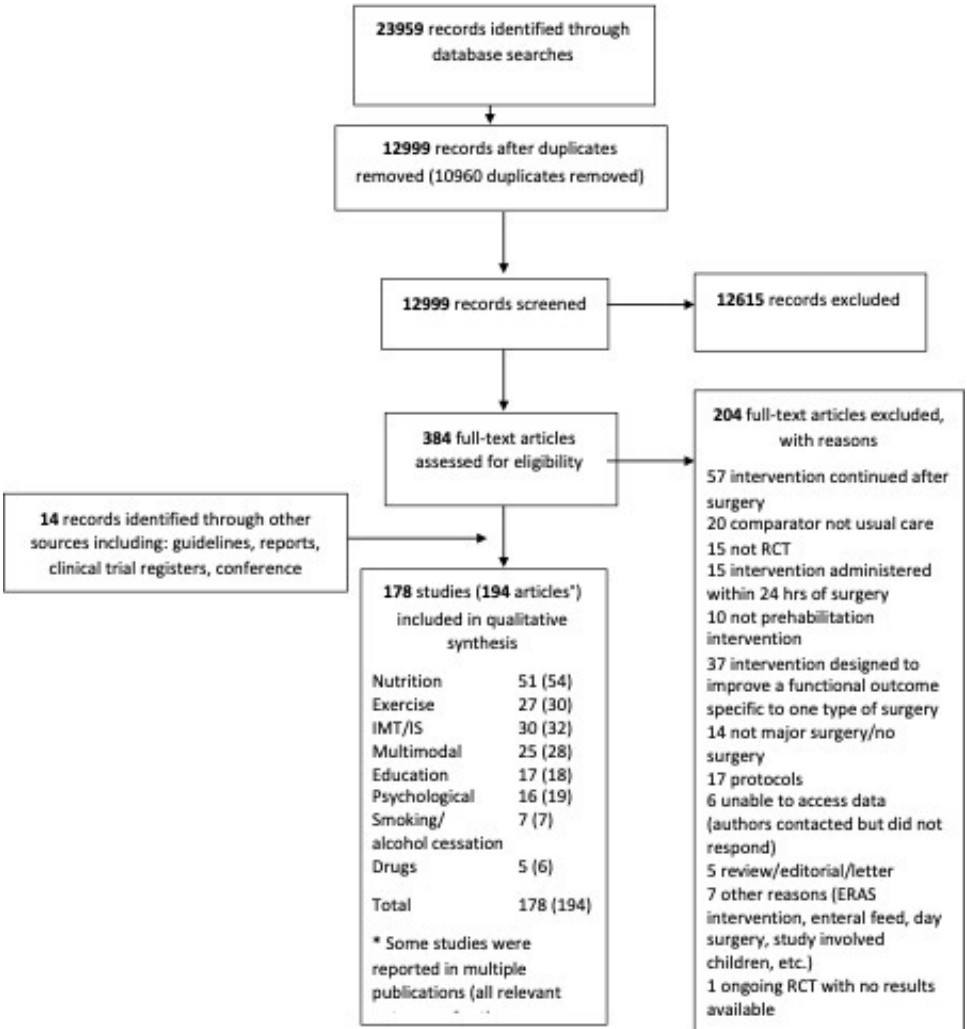

**Figure 1** PRISMA[217] flow diagram. ERAS, enhanced recovery after surgery; IMT, inspiratory muscle training; IS, incentive spirometry; PRISMA, Preferred Reporting Items for Systematic Reviews and Meta-Analyses; RCT, randomised controlled trial.

## Perioperative mortality

All-cause perioperative mortality (approximately 30 days) reported in a format usable for pooling was available from 11 of 19 immunonutrition studies[36–40 45–47 72 74 80] (910 participants); 4 of 6 ONS studies[30–33] (342 participants); 3 of 6 pre/probiotics studies[58–60] (214 participants); 5 of 26 exercise studies[112 120 121 126 128] (406 participants); 6 of 18 IMT studies[84 86 91 93 108 109] (407 participants); 3 of 16 education studies[160 166 174] (938 participants) and 10 of 25 multimodal studies[108 138 140 141 148 150 151 156 157 206] (771 participants). Mortality rates ranged from 1% to 5%. There was no effect of any prehabilitation intervention on all-cause mortality (figure 2). There was no evidence of statistical heterogeneity between studies for any intervention ($I^2$=0) other than ONS and pre/probiotics ($I^2$=33% and 32%, respectively). The GRADE quality of evidence ranged from low to very low.

## Hospital LoS

IMT (10 of 18 studies,[81 85 87 88 91–93 106 108 109] 1003 participants), immunonutrition (13 of 19 RCTs,[37–41 43 45–47 70 72 73 80] 1010 participants) and multimodal interventions (18 of 25 RCTs,[100 108 137–143 145 148 149 151–153 156 157 206] 1529 participants) reduced LoS (figure 3), by −1.81 days (95% CI −2.31 to −1.31, −2.11 days (95% CI −3.07 to −1.15) and −1.67 days (95% CI −2.31 to −1.03), respectively. Statistical heterogeneity was moderate for immunonutrition (31%) and high for multimodal ($I^2$=80%). The GRADE quality of evidence was moderate for IMT, low for multimodal and very low for immunonutrition.

## Total complications (infective and non-infective)

Results were pooled for immunonutrition (7 of 19 studies,[37 39 40 42 44 46 74] 727 participants), exercise (5 of 26 studies,[112 117 123 125 135] 287 participants) and multimodal (5 of 25 studies,[138 142 150 155 206] 313 participants) (figure 4). Multimodal interventions reduced risk of total complications by 16% (risk ratio (RR) 0.84, 95% CI 0.72 to 0.97, $I^2$=0%). The GRADE quality of evidence was very low or low for all three interventions.

## Total infective complications

Total infective complications were reported in 6 of 19 immunonutrition studies[36 38–40 43 72] (609 participants),

**Table 1** Summary of prehabilitation interventions, main results and GRADE quality of evidence rating

| Pre-admission intervention | N studies identified and study characteristics | Surgical populations included | Interventions | Main results |
|---|---|---|---|---|
| **Nutritional interventions** | | | | |
| Oral nutritional supplements)[30–35 69] | 7 studies published between 2000 and 2018, including 23–125 ppts. 4 UK 2 Japan 1 China | 2 hepatectomy 4 colorectal/GI 1 oesophagogastric junction cancer | Fortisip/Fortijuice (2 studies, 1 of which also gave dietary advice), Nutrison fibre (1 study), un-named liquid formulas/oral feeds (3 studies, 1 of which also gave dietary advice), or Livact (1 study). Where stated volumes ranged from 250 mL/day to 1400 mL/day, and duration ranged from 5 days to 1 month. Oral consumption, usually self-administered at home. 3 studies reported intervention delivered by dietitian/nutritionist. Number of contacts generally not reported. Comparator: standard diet/ no supplementation (5 studies), dietary advice (2 studies). | **Mortality** RR 1.18 (95% CI 0.23 to 6.11), p=0.85 GRADE rating: very low **LoS** MD −0.54 (95% CI −1.38 to 0.29), p=0.20 GRADE rating: very low Could not pool results for the other outcomes |
| Immunonutrition[36–47 70–75 80] | 19 studies published between 2002 and 2020, including 14–244 ppts. 5 Western Europe 4 Japan 2 Turkey 1 Australia 1 USA 1 India 1 Lithuania 1 Mexico 1 New Zealand 2 not stated | 2 colorectal cancer 1 upper or lower GI 3 GI cancer 1 total gastrectomy (cancer) 4 pancreatic cancer 2 any abdominal surgery 1 lung cancer 2 cardiac surgery 2 hepatectomy/liver cancer 1 enterocutaneous fistula | Most (15 studies) used combined arginine, omega-3 fatty acids, and RNA; 1 study used alanyl glutamine, 1 used L-glutamine, 1 used arginine +glutamine, and 1 used L-arginine +PUFA), and, where reported, generally ranged from 711 mL/day to 1 L/day, for 3–10 days. Where reported, usually oral consumption at home, hospital, or home and hospital. Comparator: no supplement/usual care/standard diet (17 studies), maltodextrin (1 study), NR (1 study). | **Mortality** RR 0.55 (95% CI 0.21 to 1.42), p=0.22 GRADE rating: low **LoS** MD −2.11 (95% CI −3.07 to −1.15), p<0.0001 GRADE rating: very low **Total complications (infective and non-infective**) RR 0.74 (95% CI 0.54 to 1.02), p=0.07 GRADE rating: very low **Total infective complications** RR 0.64 (95% CI 0.40 to 1.01), p=0.05 GRADE rating: very low **Wound infection** RR 0.71 (95% CI 0.51 to 0.99), p=0.05 GRADE rating: very low **Pneumonia** RR 0.52 (95% CI 0.18 to 1.44), p=0.21 GRADE rating: very low |
| Weight loss[48–51 53–57 76 205 218] | 11 studies published between 2007 and 2019, including 21–294 ppts. 2 UK 4 Western Europe 2 USA 1 Brazil 1 Australia 1 not stated | 7 bariatric (including Roux-en-Y gastric bypass and sleeve gastrectomy) 1 cardiac 1 nephrectomy 1 partial hepatectomy 1 general surgery (hernia repair/cholecystectomy) | Very low-calorie or low-calorie liquid diet or ppt-led calorie restriction (with/without diet sheets). Where reported, length of time on diet ranged from 3 days to 8 weeks (in 4 studies this was 14 days). All home-based interventions; some (5 studies) delivered by dietitian/nutritionist and some (6 studies) reported number of contacts (range 1–3 times or described as 'regular phone calls'). Comparator was mostly usual diet or standard care (note that standard diet in one study was 1000 kcal/day low-carbohydrate, high-protein diet). | **LoS** MD 0.22 (95% CI −0.46 to 0.91), p=0.53 GRADE rating: very low Could not pool results for the other outcomes |

Continued

| Table 1 Continued | | | | |
|---|---|---|---|---|
| **Pre-admission intervention** | **N studies identified and study characteristics** | **Surgical populations included** | **Interventions** | **Main results** |
| Pre/probiotics[58–61 77 78] | 6 studies published between 2004 and 2019, including 55–137 ppts.<br>3 Europe<br>2 Brazil<br>1 China | 5 colorectal resection/surgery for colorectal cancer or elective laparotomy (predominantly colectomy)<br>1 liver transplant | Five different formulations:<br>Probiotic capsule +prebiotic (oligofructose)<br>Lyophilised yeast capsule +*Saccharomyces boulardi*<br>ProBacti 4 Enteric (*Lactococcus lactis, Lactobacillus casei, L. acidopholous*, and *Bifidobacterium bifidum*) Simbioflora (fructooligosaccharide, *L. acidophilus* NCFM, *L. rhamnosus* HN001, *L. casei* LPC-37, and *B. lactis* HN019)<br>Synbiotic 2000 FORTE (lactic acid bacteria, *Pediacoccus pentosaceus* and *Leuconostoc mesenteroides*)<br>Oral bifid triple viable capsules (*B. longum, L. acidophilus* and *Enterococcus faecalis*)<br>Usually home-based oral intervention for 3 or 7 days (not stated in two studies). 2 studies reported daily phone calls (number of contacts not stated for others).<br>Comparator was placebo capsules/powder (4 studies) or usual care (2 studies). | **Mortality**<br>RR=0.76 (95% CI 0.17 to 3.42), p=0.72<br>GRADE rating: low<br>**Total PO infective complications**<br>RR 0.48 (95% CI 0.14 to 1.62), p=0.23<br>GRADE rating: low<br>Could not pool results for the other outcomes |
| Nutritional optimisation[62–64] | 3 studies published between 1987 and 2014 including 35–41 ppts.<br>1 UK<br>1 USA<br>1 not stated | 1 Roux-en-Y gastric bypass<br>1 upper GI<br>1 elective CABG | Optimisation of glucose or general nutrition, or low glycaemic index diet. Duration ranged from 10 days to 3 months. Where reported, generally oral/written advice (one or two contacts), self-delivered at home.<br>Comparator was nutrition counselling, or high glycaemic index diet (not stated in one study). | No meta-analyses conducted on any outcome due to limited data |
| Other nutritional interventions[65–68 79] | 5 studies published between 2007 and 2019, including 30–105 ppts.<br>2 Europe<br>1 India<br>1 Australia<br>1 Iran | 3 CABG/cardiac<br>1 lung cancer<br>1 lumbar spine | Five different interventions:<br>Combined supplement (glutamine, L-carnitine, vitamins C, E, and selenium)<br>Vitamin D<br>Coenzyme Q10<br>a-ketoglutaric acid and 5-hydroxymethylfurfural<br>Fish oil<br>Oral consumption, usually at home, for 7 days–5 weeks. Comparator was placebo (2 studies) or no supplement/usual nutrition/usual care (3 studies)<br>None reported the number of contacts. | No meta-analyses conducted on any outcome due to limited data and because of intervention heterogeneity |
| Exercise[110–126 128 129 131–136 219] | 27 studies published between 1996 and 2020, including 14–164 ppts.<br>1 Russia<br>7 UK<br>1 Ireland<br>6 Western Europe<br>3 Canada<br>2 Australia<br>2 USA<br>1 Japan<br>1 Brazil<br>3 not stated | 1 cardiac<br>1 radical cystectomy<br>2 AAA<br>1 bariatric<br>6 TKR/THR//THA/TKA<br>2 liver resection<br>3 lung cancer<br>1 radical prostatectomy<br>1 abdominal surgery<br>2 lumbar spine surgery<br>1 thoracic surgery<br>1 urological surgery<br>1 rectal cancer<br>2 colorectal surgery<br>1 cardiac or thoracic surgery<br>1 knee osteoarthritis | Different protocols involving different amounts of cardio and strengthening exercises in both group and individual format. Individual programmes often tailored, for example, physiotherapy. Many studies reported supervision. Specific intensity sometimes mentioned, eg, moderate, HIIT. Length of intervention: 26 studies: 1–8 weeks; 1 study: 15–17 weeks. Comparator: usual care (22 studies), usual care and diet therapy (1 study), usual care and additional exercise regimen (1 study), NR (2 studies). | **Mortality** RR 0.74 (95% CI 0.23 to 2.35), p=0.61<br>GRADE rating: low<br>**LoS** MD −0.38 days (95% CI −0.82 to 0.06), p=0.09<br>GRADE rating: very low<br>**Total PO complications**<br>RR 0.83 (95% CI 0.61 to 1.12), p=0.22<br>GRADE rating: low<br>**Pneumonia**<br>RR 0.72 (95% CI 0.35 to 1.44), p=0.35<br>GRADE rating: very low<br>**PPCs** RR 0.54 (95% CI 0.39 to 0.75), p=0.0003<br>GRADE rating: low |

Continued

**Table 1** Continued

| Pre-admission intervention | N studies identified and study characteristics | Surgical populations included | Interventions | Main results |
|---|---|---|---|---|
| Inspiratory muscle training (IMT)[81–88 90–95 106 108 109 220] | 18 studies published between 1996 and 2020, including 16–279 ppts. 7 Western Europe 1 UK 1 Eastern Europe 4 Brazil 3 China 1 Israel 1 not stated | 7 cardiac 3 thoracic (lung cancer) 1 abdominal aortic aneurysm 1 bariatric 1 laparoscopic bariatric 1 colorectal cancer 1 THR 1 abdominal or urological 2 oesophagectomy | Threshold IMT starting at 30%–40% MIP 4×/day to 2×/week (but most at least once daily) for 4 days–4 weeks before surgery (most for 2 weeks). Most included weekly contact with ppts. Comparator usual care; only 1 study included a sham IMT training group as a comparator. | **Mortality** RR 1.49 (95% CI 0.60 to 3.69), p=0.39 GRADE rating: low **LoS** MD −1.81 days (95% CI −2.31 to −1.32), p<0.00001 GRADE rating: moderate **PPCs** RR 0.55 (95% CI 0.38 to 0.80), p=0.002 GRADE rating: low **Pneumonia** RR 0.69 (95% CI 0.49 to 1.05), p=0.08 GRADE rating: very low |
| Incentive spirometry (IS)[90 96–99] | 5 studies published between 1983 and 2014, including 41–172 ppts. 2 USA 1 UK 1 Brazil 1 not stated | 1 cardiac 1 THR 1 abdominal (type not stated) 1 laparoscopic (bariatric) | IS (different protocols, generally 4–10 repetitions/day) for 1 week before surgery. Comparator usual care; 1 study included sham IS. | **LoS** MD −2.39 (95% CI −5.50 to 0.72), p=0.13 GRADE rating: very low **PPCs** RR 0.68 (95% CI 0.25 to 1.81), p=0.44 GRADE rating: very low No meta-analyses conducted on any other outcome due to limited data |
| Combined respiratory interventions[100–105 107] | 7 studies published between 1998 and 2018, including 9–60 ppts. 3 Spain 1 USA 1 Turkey 2 not stated | 1 thoracic (lung cancer) 2 cardiac (ppts with COPD) 1 oesophagectomy 3 laparoscopic bariatric | Respiratory rehabilitation (multimodal intervention designed for people with impaired lung function; includes physical exercises, breathing exercises, education, etc) for 1–4 weeks before surgery. Chest physiotherapy programmes (combinations of IS, IMT and lung re-expansion). Comparator usual care. | No meta-analyses conducted on any outcome due to limited data |
| Combined interventions[100 108 137–159] | 25 studies published between 2000 and 2019 including 14–249 ppts. 3 Western Europe 8 USA/Canada 5 China/Taiwan 1 Turkey 1 Egypt 1 Australia 1 Hungary 5 not stated | 8 lung resection 3 CABG 2 oesophagogastric resection 1 pancreaticoduodenectomy 2 colorectal cancer 1 prostatectomy 1 cystectomy 4 THR/TKR 2 elective abdominal surgery 1 cardiac/thoracic | Interventions combined 2–4 different modes. All studies included physical activity, 10 included breathing exercises, 5 education, 7 nutrition, 8 psychological and 1 drug optimisation. The most common combinations were physical activity +breathing (6 studies) and physical activity +education (5 studies). Comparator usual care; 1 study used a standard exercise +nutrition protocol and a standard nutrition protocol. | **Mortality** RR 0.67 (95% CI 0.23 to 1.95), p=0.46 GRADE rating: low **LoS** MD −1.67 days (95% CI −2.31 to −1.03), p<0.00001 GRADE rating: moderate **Pneumonia** RR 0.56 (95% CI 0.28 to 1.12), p=0.10 GRADE rating: very low **Total PO complications** RR 0.84 (95% CI 0.72 to 0.97), p=0.02 GRADE rating: very low |
| Education[160–167 169–176 221] | 17 studies published between 1996 and 2020 including 35–441 ppts. 6 USA/Canada 4 Western Europe 2 Finland 2 Australia 1 Serbia 2 Turkey | 4 cardiac 1 laparoscopic cholecystectomy 1 unspecified abdominal 1 spinal surgery 8 THR/TKR 1 arthroscopic rotor cuff repair 1 unspecified | Structured education of different levels of intensity delivered through interviews and written information (7 studies), classes and written information (4 studies), interview and website (1 study), education session only (1 study), booklet, interview and telephone call (1 study), DVD and telephone call (1 study), telephone call only (1 study), education booklet only (3 studies). Education consisted of information on what to expect from surgery, teaching exercises for use postoperatively and the use of aids. Comparator usual preoperative care and explanations (delivered verbally, in written form and via video). 1 study compared intensive education sessions with practical classes with a physiotherapist. | **Mortality** RR 0.83 (95% CI 0.24 to 2.95), p=0.78 GRADE rating: low **LoS** MD 0.00 days (95% CI −0.40 to 0.40), p=1.00 GRADE rating: very low No meta-analyses conducted on any other outcome due to limited data |

Continued

**Table 1** Continued

| Pre-admission intervention | N studies identified and study characteristics | Surgical populations included | Interventions | Main results |
|---|---|---|---|---|
| Psychological[177–192] | 16 studies published between 1986 and 2020 including 24–400 ppts.<br>8 Western Europe<br>2 US<br>1 China<br>1 Nigeria<br>1 Pakistan<br>1 Columbia<br>2 not stated | 3 cardiac (one with additional poorly controlled risk factor)<br>2 abdominal (type not stated)<br>1 radical prostatectomy<br>1 colorectal cancer<br>1 cholecystectomy<br>1 bariatric<br>2 lumbar fusion<br>1 general surgery<br>1 pancreatic surgery<br>1 knee replacement surgery<br>2 varied or unspecified elective surgery | Different protocols involving psychological therapies: expectation management; relaxation exercises; breathing exercises; guided imagery; mindfulness; stress management; counselling; 7 studies involved cognitive–behavioural therapy. 4 stated that a psychologist delivered the intervention. Comparator majority were usual care; 1 study used a control topic; 3 studies gave information about surgical procedures or general advice; 1 study offered a hospital helpline number; 1 study an information session with ppt and support person. | **LoS** MD −0.82 days (95% CI −1.83 to 0.19), p=0.11<br>GRADE rating: very low<br>No meta-analyses conducted on any other outcome due to limited data |
| Smoking cessation[193–197] | 5 studies published between 2004 and 2014, including 28–168 ppts.<br>2 Western Europe<br>1 USA/Canada<br>1 Australia<br>1 China | 1 lower or upper fracture<br>1 THR<br>1 not stated<br>2 general surgery | Majority used a combination of medication (nicotine replacement/bupropion) and/or advice giving/counselling (face-to-face or telephone) for 4–8 weeks before surgery.<br>Comparator usual care. | **Wound infection**<br>RR=0.28 (95% CI 0.12 to 0.64), p=0.002<br>GRADE rating: very low |
| Alcohol cessation[199 222] | 2 studies published in 1999 and 2002, including 42 and 28 ppts., respectively<br>Both Western Europe | 1 colorectal<br>1 hip arthroplasty | Withdrawal from alcohol, motivational counselling and treatment with disulfiram for 1–3 months before surgery.<br>Comparator usual care. | No meta-analyses conducted on any outcome due to limited data |
| Pharmacological[200–204] | 5 studies published between 2005 and 2020, including between 4 and 400 ppts.<br>3 Western Europe<br>1 Iran<br>I Australia | 2 CABG<br>1 cardiac surgery<br>1 prostatectomy<br>1 major non-cardiac surgery | Intervention:<br>Provision of drugs (serenoa repens (Permixon); atorvastatin; bisoprolol titration; allopurinol and vitamin E supplement); optimising dosage; management and control of risk factors; 1 nephrologist management; 1 nurse-led strategy.<br>Duration of intervention: 3–5 days— 2 months before surgery.<br>Comparator: 4 usual care; 1 placebo | No meta-analyses conducted on any outcome due to limited data |

CABG, coronary artery bypass graft; COPD, chronic obstructive pulmonary disease; GI, gastrointestinal; GRADE, Grading of Recommendations, Assessment, Development and Evaluation; HIIT, high intensity interval training; LoS, length of stay; MD, mean difference; MIP, maximum inspiratory pressure; NR, not reported; PO, postoperative; PPCs, postoperative pulmonary complications; ppts, participants; PUFA, polyunsaturated fatty acids; RR, risk ratio; THA, total hip arthroplasty; THR, total hip replacement; TKA, total knee arthroplasty; TKR, total knee replacement.

and 3 of 6 pre/probiotic studies[58–60] (214 participants). Immunonutrition and pre/probiotic interventions reduced the risk of infective complications by 36% (RR 0.64, 95% CI 0.40 to 1.01, $I^2$=56%) and 52% (RR 0.48, 95% CI 0.14 to 1.62, $I^2$=66%), respectively. The GRADE

| | Number of studies | Intervention (n/N) | Control (n/N) | Risk ratio (95% CI) | P value | $I^2$ (%) |
|---|---|---|---|---|---|---|
| Immunonutrition | 11 | 5/453 | 11/457 | 0.55 (0.21, 1.43) | .22 | 0 |
| Multimodal | 10 | 5/383 | 8/388 | 0.67 (0.23, 1.95) | .46 | 0 |
| Exercise | 5 | 5/202 | 7/204 | 0.74 (0.23, 2.37) | .61 | 0 |
| Probiotics | 5 | 9/174 | 9/170 | 0.76 (0.17, 3.41) | .72 | 32 |
| Education | 3 | 4/470 | 5/468 | 0.83 (0.24, 2.91) | .78 | 0 |
| ONS | 4 | 6/172 | 5/170 | 1.18 (0.23, 6.08) | .85 | 33 |
| IMT | 6 | 11/205 | 7/202 | 1.49 (0.60, 3.70) | .39 | 0 |

.15   .5   1   2   5   10
Favours intervention   Favours control

**Figure 2** Forest plot of prehabilitation for reducing all-cause perioperative mortality. All interventions were tested with usual care as control. IMT, inspiratory muscle training; ONS, oral nutritional supplements.

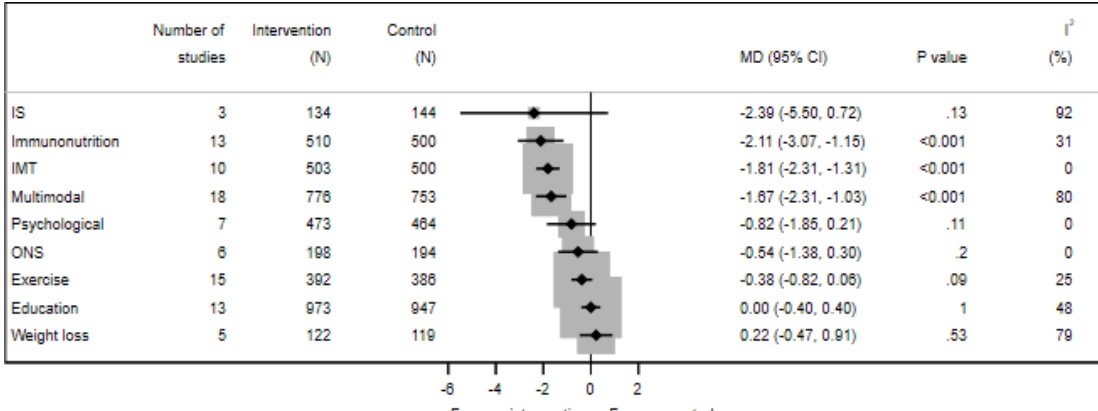

**Figure 3** Forest plot of prehabilitation for reducing length of hospital stay. All interventions were tested with usual care as control. IMT, inspiratory muscle training; IS, incentive spirometry; MD, mean difference; ONS, oral nutritional supplements.

quality of evidence was very low/low, respectively (see online supplementary digital content for plots).

### Wound infection

Wound infection was reported in 8 of 19 immunonutrition studies[36 37 39–41 70 73 80] (752 participants) and 3 of 4 smoking cessation studies[195–197] (236 participants). Immunonutrition reduced wound infection by 29% (RR 0.71, 95% CI 0.51 to 0.99, $I^2$=6%). Smoking cessation reduced wound infection by 72% (RR 0.28, 95% CI 0.12 to 0.64, $I^2$=12%). The GRADE quality of evidence was very low for both interventions (see online supplementary digital content for plots).

### Postoperative pulmonary complications

Postoperative pulmonary complications (PPCs) were reported in 5 of 18 IMT studies[81 85 87 95 106] (633 participants), 4 of 5 IS studies[96–99] (315 participants) and 4 of 26 exercise studies[112 118 121 126] (325 participants). IMT and exercise interventions reduced PPCs by 45% and 46% (RR 0.55, 95% CI 0.38 to 0.80, $I^2$=18% and RR 0.54, 95% CI 0.39 to 0.75, $I^2$=0%), respectively (figure 5). GRADE quality of evidence was low for exercise and IMT and very low for IS.

### Pneumonia

Pneumonia was reported in 11 of 18 IMT studies[82 85–88 90 91 93 95 106 108] (1052 participants), 7 of 19 immunonutrition studies[36 39 41 42 70 72 80] (521 participants), 4 of 26 exercise studies[111 121 126 207] (266 participants) and 7 of 20 multimodal studies[100 108 137 148 149 151 157 208] (341

participants) (figure 6). IMT reduced the risk of pneumonia by 31% (RR 0.69, 95% CI 0.46 to 1.04, $I^2$=18%). The GRADE quality of evidence for pneumonia was very low for all interventions.

### Funnel plots

Funnel plots were constructed for immunonutrition, IMT, multimodal, exercise and educational interventions for LoS only, as these were the only outcome that had >10 studies contributing to the meta-analyses for each intervention. None of these funnel plots showed marked asymmetry (see online supplemental digital content). The Egger's regression tests confirmed that there was no marked asymmetry for any intervention (p<0.05), except for multimodal (p=0.01), suggesting that for most of the results reporting biases (including publication bias) are unlikely to be an issue. However, for some interventions (eg, immunonutrition), the low power and heterogeneity of the included studies may limit the conclusion that can be drawn from the funnel plots.[209] We did not construct a funnel plot for immunonutrition and the mortality outcome because although we had 10 studies contributing data to this outcome, 4 of these included zero events.

### Sensitivity analyses

None of the sensitivity analyses conducted materially altered the results of the main analyses for any intervention or outcome, except for immunonutrition and the LoS outcome. When including only RCTs that had low RoB for random sequence generation and allocation

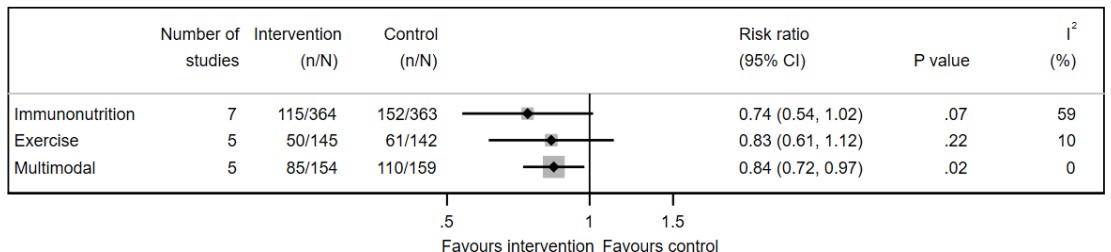

**Figure 4** Forest plot of prehabilitation for reducing total postoperative complications. All interventions were tested with usual care as control.

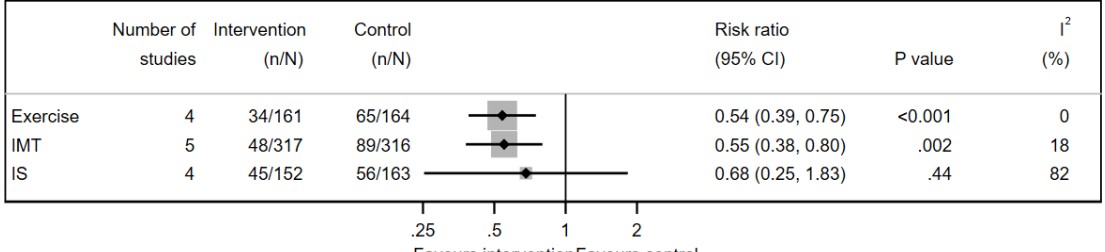

**Figure 5** Forest plot of prehabilitation for reducing postoperative pulmonary complications . All interventions were tested with usual care as control. IMT, inspiratory muscle training; IS, incentive spirometry.

concealment, the pooled estimate was attenuated (immunonutrition: –1.14 days, 95% CI –2.69 to 0.41, p=0.15). Estimates from all fixed-effects meta-analyses were similar to those from the random-effects meta-analyses. All sensitivity analyses are shown in the online supplemental digital content.

### Subgroup analyses

Of the five prespecified subgroup analyses, we pooled data for type of surgery. Although none of the studies reported having ERAS protocols as part of usual care, we performed a subgroup analysis by grouping studies published before and after 2010, when ERAS programmes began to be implemented. We could not do subgroup analysis for the other three prespecified subgroups.

Subgroup analyses by type of surgery were conducted for cancer surgery (immunonutrition, exercise and multimodal interventions), orthopaedic surgery (multimodal interventions) and cardiac surgery (IMT). In people undergoing *cancer surgery*, immunonutrition[38–41 43 80] and multimodal interventions[100 138 142 148 149 151 206 208] reduced LoS by about 2 days (mean difference (MD) –1.83, 95% CI –2.85 to –0.80) and 2.5 days (MD –2.50 days, 95% CI –4.05 to –0.95), respectively, while exercise interventions[111 120 121 125 135] did not reduce LoS (MD –0.16 days, 95% CI –0.63 to 0.31). In people undergoing *knee/hip replacement surgery*, multimodal interventions[141 143 145] reduced LoS by about 1 day (MD –1.43, 95% CI –2.84 to –0.02).

In people undergoing *cardiac surgery*, IMT[85 87 88 92 106] reduced LoS (MD –1.73 days, 95% CI –2.39 to –1.07) and pneumonia[82 86–88 95] (RR 0.53, 95% CI 0.39 to 0.73). None of the interventions reduced mortality in the above surgical subgroups.

Subgroup analyses including studies published before or after 2010 did not change any of our findings; effect size and direction were similar for all the interventions and outcomes that could be grouped, although some did not reach statistical significance because of small sample size. All subgroup analyses are shown in the online supplemental digital content.

### DISCUSSION

The main findings from this review are that four types of prehabilitation—IMT, exercise, immunonutrition and multimodal—reduced postoperative complications and/or hospital LoS. Immunonutrition reduced total infective complications by 37%, while IMT and exercise reduced PPCs by 45% and 46%, respectively. LoS was reduced by 1.5–2 days on average.

Generally, these results were robust to sensitivity analyses. For immunonutrition, the results were attenuated after removing the studies at high RoB (LoS) and studies conducted prior to 2010 (total infective complications) (see online supplemental material). The overall quality of the evidence was low. For most interventions, overall pooled sample sizes were small and CIs were wide, suggesting that many of the analyses were underpowered. Overall, prehabilitation interventions appear safe; none of the studies reported adverse effects of any of the interventions administered and most studies achieved good follow-up and reported little or no attrition.

We found no evidence of subgroup effects by type of surgery. Despite the clinical heterogeneity in the studies identified (different surgical populations and differences between some interventions), statistical heterogeneity

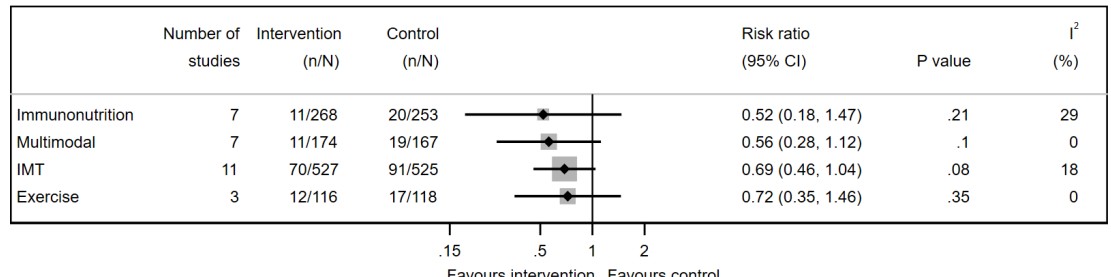

**Figure 6** Forest plot of pre-admission interventions for reducing pneumonia. All interventions were tested with usual care as control. IMT, inspiratory muscle training.

for most interventions and outcomes analysed was low, suggesting that our approach of pooling data across all surgical populations was justified. This is not surprising, given that the modifiable risk factors that reduce fitness for surgery and delay recovery are similar in different surgical populations. These risk factors include, for example, physical inactivity and low aerobic fitness, which affects between 33% and 45% of surgical populations,[210] excessive alcohol consumption and smoking, which affects about a quarter,[211] and obesity, which affects one-third.[212]

There was marked clinical heterogeneity for some interventions, with variations in their components, duration and mode of administration; for example, educational interventions ranged from written information (a booklet) that patients were sent home with to one-to-one structured education programmes. Similarly, psychological interventions ranged from written instructions on relaxation exercises and deep breathing to cognitive–behavioural therapy sessions with a psychologist.

### Limitations of the evidence

The trials that contributed data to these meta-analyses were largely of low or very low quality. Most trials were small (IMT 20–200 participants; immunonutrition 14–244 participants; exercise 14–164 participants) and not blinded, since blinding in trials of lifestyle interventions is challenging and often impossible. A systematic review of 45 trials reported that higher expectations of recovery positively influenced patient outcomes.[213] Thus, the apparent intervention effect may be influenced by patient expectations rather than the active intervention. Nevertheless, IMT, exercise and immunonutrition trials all improved physiological markers of either exercise capacity (eg, inspiratory muscle strength, 6-minute walk test) or biomarkers of immune function, and these parameters remained unchanged in the usual care groups, which suggests that the interventions had physiological effects. The influence of blinding on effect size in trials is not clear cut; for example, a recent meta-epidemiological study in physical therapy trials found no relationship between lack of blinding and effect size, and, surprisingly, trials with inadequate blinding tended to underestimate treatment effects.[214] A further limitation for the immunonutrition trials was that they were mostly industry sponsored and there is evidence that industry-sponsored studies are biased in favour of the sponsor's products.[209]

Another limitation is the lack of standardised definitions of postoperative complications (infective, non-infective and pulmonary complications). Many studies reported individual complications without reporting total complications; therefore, we could not include them in meta-analyses. Similarly, LoS data were inconsistently reported (mean and SD or median and range or median and IQR) and in some cases no variance data were reported, so these studies could not be included in the meta-analysis. LoS is one of the main clinical outcomes of interest in

this research area; however, it is likely to be influenced by variation in discharge criteria, which may result in differences between studies.

We had no information on whether studies were conducted pre-ERAS or post-ERAS implementation, and there is some evidence that beneficial effects of interventions carried out before surgery disappear when ERAS is introduced.[215] However, our sensitivity analysis restricting to the previous 10 years (2010–2020), when most ERAS programmes were introduced, did not change effect sizes for any of the interventions and outcomes investigated.

Timing of mortality assessment within hospitals was not always clearly defined or consistent between studies. This lack of uniformity is likely to have caused rates of reporting to differ between studies. Further variability may have been introduced as definitions of some of our secondary outcomes of interest were not always clear and often differed between studies. An adequate description of the comparator (largely usual care) was absent from most study reports; it was therefore difficult to determine what usual care was, whether there were any enhanced recovery protocols in place, or even whether components were added to usual care for the purpose of the trial.

Few of the studies we identified reported outcomes beyond 30 days, therefore the effect of prehabilitation on longer term outcomes such as hospital readmission and mortality are not known. Also, the extent to which behaviour change in the preoperative period was maintained postoperatively, or whether this behaviour change leads to change in modifiable risk factors, remains to be assessed. Few studies (apart from the psychological intervention studies) included patient-reported outcomes such as QoL, pain, satisfaction with pre-admission intervention and care and factors associated with mental health. Finally, all interventions were initiated and followed up in hospital, with little consideration for how primary healthcare services could be integrated into the patient pathways for continuity of care after surgery.

### Strengths and limitations of the review

A major strength of the review is that it provides a summary of the collective evidence on prehabilitation for all surgical patients. The systematic methods employed to identify the included studies were stringent, with inclusion of published literature in all languages, alongside grey literature searching, to avoid publication bias.

A limitation is that by having a stringent definition of prehabilitation as an intervention occurring only in the preoperative period, we excluded a lot of studies where the intervention continued postoperatively. This excluded 5 smoking cessation studies, in which the smoking cessation intervention almost always continues postoperatively, as well as 18 nutrition, 4 exercise, 6 respiratory, 1 multimodal, 3 education, 1 psychological, and 3 pharmacological intervention studies. A final limitation is the potential for effect size multiplicity[216] (multiple dependent effect sizes, for example, in related outcomes such as infective complications, wound infection and pneumonia, derived

from the same individual participants). We chose to perform separate analyses for each of these outcomes rather than averaging multiple effect sizes within studies because this is what we prespecified in our protocol and because we did not consider effect sizes for each outcome to be equivalent. We acknowledge the risk of inflating type 1 error rates.

### Deviations from the protocol

We could not complete two of the subgroup analyses that we had prespecified (by differences in intervention characteristics and high vs low surgical risk patients) because of a lack of data. We also decided not to attempt to compare the effectiveness of the different components given the large variability within each intervention (in mode/place of administration, intensity and duration of intervention, etc).

### Agreements and disagreements with other research

Despite the broad scope of this review, the inclusion of multiple surgical populations and the exclusion of studies in which the interventions were continued postoperatively, our results are similar to those reported for the specific prehabilitation interventions in single surgical populations.

### Future research/clinical recommendations

Our review has highlighted that IMT, exercise, immunonutrition and smoking cessation interventions should be considered as part of multimodal prehabilitation programmes. Further research is needed on how best to identify the high-risk patients who are most likely to benefit from the various components of a multimodal prehabilitation intervention because treatment needs to be individualised, taking into account patient need, preferences and likelihood of adherence to the different components. For example, IMT could be an adjunct to exercise or replace exercise in those with impaired respiratory function, while a psychological intervention should only be offered to those who are likely to benefit, while a nutritional intervention needs to be tailored to whether the patient is malnourished or underweight or overweight. A well-designed, large, pragmatic, multicentre clinical trial is needed to determine the true effectiveness of an individualised multimodal intervention. Such a trial should collect long-term outcome data and patient-reported outcomes data, including outcomes related to mental health. It should also measure adherence to the different components of the intervention and longer term behavioural/lifestyle changes. It should also investigate the mechanisms through which the different components of prehabilitation work. The impact of shorter LoS on the broader health and social care system and on long-term patient outcomes should also be considered. Future trials should also employ digital technology to monitor adherence and provide feedback to patients and also include aspects of implementation and scaling up of the interventions in the National Health Service (NHS).

### CONCLUSIONS

Some prehabilitation interventions, in particular IMT and immunonutrition, may reduce hospital LoS and some postoperative complications. Overall, the quality of the evidence was low or very low. Despite the relatively large number of studies identified, most had very small sample sizes and our pooled analyses were likely underpowered.

**Author affiliations**
[1]NIHR Bristol BRC, University Hospitals Bristol and Weston NHS Foundation Trust, University of Bristol, Bristol, UK
[2]Centre for Exercise, Nutrition and Health Sciences, University of Bristol, Bristol, UK
[3]Population Health Sciences, Bristol Medical School, University of Bristol, Bristol, UK
[4]Bristol Trials Centre (CTEU), Bristol Medical School, University of Bristol, Bristol, UK
[5]School of Oral and Dental Science, University of Bristol, Bristol, UK
[6]Faculty of Health, Victoria University of Wellington, Wellington, New Zealand
[7]University Hospitals Bristol and Weston NHS Foundation Trust, University of Bristol, Bristol, UK

**Acknowledgements** We thank Alison Richards and Sarah Dawson for conducting the literature searches, Zoe Zou and Sarah Sauchelli Toran for translating texts into English, Shirley Jenkins and Sofia Leadbetter for help with referencing and Alex Whitmarsh for statistical advice. The following contributed to abstract screening or data extraction: Sharea Ijaz, Lauren Scott, Agnieszka Skorko, George Snell, Helena Wilkes, Mariella Williams and Annabelle Johnson. We also thank Professor Julian Higgins for general advice on systematic review conduct and reporting.

**Contributors** RP conducted the study (screened abstracts and full texts, extracted data, assessed risk of bias, conducted GRADE assessmet, conducted all analyses and assembled tables of results) and wrote the first draft of the manuscript. GH conducted the study (extracted data, assessed risk of bias, completed handsearching, assembled tables of results, checked analysis, conducted GRADE assessment). CA and CE conducted the study (extracted data, completed handsearching and assembled tables of results, checked analysis and formatting, conducted GRADE assessment). KN, SB, TB, AC and AH extracted data. JH conducted some of the statistical analyses (Egger's regression tests) and constructed all the plots in the main manuscript. AN critically reviewed the manuscript. SS had the idea for conducting a study of prehabilitation in patients undergoing major elective surgery and provided clinical input. MP designed and conducted the study, interpreted the data and wrote the manuscript. The manuscript's guarantors (MP and RP) affirm that the manuscript is an honest, accurate, and transparent account of the study being reported; that no important aspects of the study have been omitted; and that any discrepancies from the study as planned (and, if relevant, registered) have been explained.

**Funding** This work was supported by the Elizabeth Blackwell Institute (University of Bristol) and the Bristol National Institutes of Health Research (NIHR) Biomedical Research Centre.

**Competing interests** None declared.

**Patient consent for publication** Not applicable.

**Provenance and peer review** Not commissioned; externally peer reviewed.

**Data availability statement** Data are available upon reasonable request. "The aggregate data for all the meta-analyses that were conducted are listed in in the Supplementary Digital Content. These data were entered into RevMan and can be made available as an XML file upon request from the corresponding author.".

**ORCID iD**

Rachel Perry http://orcid.org/0000-0001-5874-3016

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
