## [Reviewer comments · BMJ Open]

This paper was submitted to a another journal from BMJ but declined for publication following peer review. The authors addressed the reviewers' comments and submitted the revised paper to BMJ Open. The paper was subsequently accepted for publication at BMJ Open.

ARTICLE DETAILS

TITLE (PROVISIONAL)	Pre-admission interventions (prehabilitation) to improve outcome after major elective surgery: A systematic review and meta-analysis
AUTHORS	Perry, Rachel; Herbert, Georgia; Atkinson, Charlotte; England, Clare; Northstone, Kate; Baos, Sarah; Brush, Tim; Chong, Amanda; Ness, Andy; Harris, Jessica; Haase, Anne; Shah, Sanjoy; Pufulete, Maria

VERSION 1 - REVIEW

REVIEWER	Dunne, Declan University Hospital Aintree, Liverpool Hepatobiliary Centre
REVIEW RETURNED	16-Jun-2020

GENERAL COMMENTS	Aim: To determine the benefits and harms of pre-admission interventions (prehabilitation) on postoperative outcomes in patients undergoing major elective surgery, using a meta-analysis of RCT. Summary: This is an expansive project, running to 275 pages (longer than my thesis on prehabilitation). Knowing the topic well from my higher degree, I would say the findings were as I would have expected. I think more should be made of the effect of multimodal programs, which reduced LOS, and complications in this study. They were not mentioned in the summary document which will make the paper edition of the BMJ, and I think this is an error to omit given the findings. Multimodal programs of prehabilitation will undoubtedly be the future for prehabilitation programs, and this very useful paper helps inform the key question which is – what is the best multimodal prehabilitation program. From this it would suggest something along the lines of – Exercise, with IMT and immunonutrition as a core with other aspects bolted on. There are some concerns I have which are outlined below. Major Concerns 1. I have some concerns with paragraph three. It feels like a paragraph trying to justify the need for this paper, This is undoubtedly the most expansive prehabilitation review undertaken. I am not sure it adds much to the paper.2. The first paragraph in the discussion states what your methods have done? Normally it this would be the summary paragraph for the key message and findings of a paper?3. I would rewrite the second paragraph of the discussion as the use of bullet points it makes it challenging to read, and interrupts the flow.4. Third paragraph of discussion, I think you should cautious
---

	reporting on non-statistical directions of effect, these are not statistically significant. It could lead to nervousness about IMT for example given the unexpected and unexplainable direction of travel towards increasing mortality (whilst cutting LOS and complications) 5. I think my main concern is that we do not highlight the multimodal affect. I think this is probably the future (as it was in ERAS 20 years ago), this is the key concept going forward. Minor Concerns 1) 15 authors seems a lot to be listed for a meta-analysis – are they all justified? Could some be acknowledged rather than co-authors? (I am ok if not but it seems expensive) 2) The word count seems quite high – could this be cut significantly (aprox 2500-3000 words is typical for a paper, though I accept that for a project as expansive as this it may be higher. If the paper word count could be cut it would improve the way it reads. Perhaps more in the supplementary section? 3) First paragraph – saying up to 75% of patients undergoing major surgery is a negative view. I would say a typical rate (I tend to quote 40-50%) rather than the worst possible view. I accept why it was put but it increases fear about elective surgery which I am not sure is warranted. 4) Second paragraph – do we need to mention COVID? is this specifically relevant to the paper? 5) Third paragraph of introduction – please delete the bit – it’s not needed. “e.g. exercise in cancer surgery 22; weight loss interventions in bariatric surgery 23; inspiratory muscle training in cardiac surgery 24; exercise in gastrointestinal cancer surgery 25 26, immunonutrition in head and neck surgery 27, etc.” 6) Typically robotic radical prostatectomy is done one day, and the patient discharged the next, were these studies excluded? 7) Methods are generally ok, but quite wordy. Similarly for the results. 8) Again COVID mentioned in discussion – is this appropriate? 9) The limitations section is very long – can you limit this?
--	---

REVIEWER	Reviewer 2
REVIEW RETURNED	18-Jun-2020

GENERAL COMMENTS	This is a large systematic review and meta-analysis including 144 randomized controlled trial to determine the risks and benefits of pre-admission interventions on postoperative outcomes in patients undergoing elective surgical procedures. The results demonstrate that some pre-admission interventions may reduce hospital stay and some postoperative complications. First of all, I would like to appreciate the authors extensive work! Although of high interest, there are some major points which drew my attention.  - Trial selection includes trials from 1984 to 2019. The authors already state this as limitation of the trial. Why did the authors decide to include old trials in this analysis? Perioperative treatment has been changed in the last decades which impact patients' outcomes. I think this impedes the explanatory power of the results. - The analysis includes palliative surgical procedures. Please clarify this. Does this mean cancer surgery in general or really palliative surgical procedures. If palliative surgical procedures in a narrow sense, did the authors perform a subgroup analysis (palliative/non-palliative) to see whether this may distort the mortality results? - Most of the evidence grades demonstrate very low to low GRADE
---

	levels. Why did the authors include so many low evidence studies instead of focusing on larger trials with higher evidence levels? - I would recommend to rewrite the results section. From my perspective, it is hard to read as in every paragraph details on where the information comes from are provided. I would recommend to more clearly present the results. - Please liaise with the policies of the journal, e.g., the discussion section is approximately 6 pages long.
--	--

VERSION 1 – AUTHOR RESPONSE

Reviewer: 1

Comments:

Reviewer – Declan Dunne

Aim: To determine the benefits and harms of pre-admission interventions (prehabilitation) on postoperative outcomes in patients undergoing major elective surgery, using a meta-analysis of RCT.

Summary: This is an expansive project, running to 275 pages (longer than my thesis on prehabilitation). Knowing the topic well from my higher degree, I would say the findings were as I would have expected. I think more should be made of the effect of multimodal programs, which reduced LOS, and complications in this study. They were not mentioned in the summary document which will make the paper edition of the BMJ, and I think this is an error to omit given the findings. Multimodal programs of prehabilitation will undoubtedly be the future for prehabilitation programs, and this very useful paper helps inform the key question which is – what is the best multimodal prehabilitation program. From this it would suggest something along the lines of – Exercise, with IMT and immuno-nutrition as a core with other aspects bolted on. There are some concerns I have which are outlined below.

Major Concerns

1. I have some concerns with paragraph three. It feels like a paragraph trying to justify the need for this paper, This is undoubtedly the most expansive prehabilitation review undertaken. I am not sure it adds much to the paper.

Removed.

2. The first paragraph in the discussion states what your methods have done? Normally it this would be the summary paragraph for the key message and findings of a paper?

Removed.

3. I would rewrite the second paragraph of the discussion as the use of bullet points it makes it challenging to read, and interrupts the flow.

Removed the bullet points and rewritten.

4. Third paragraph of discussion, I think you should cautious reporting on non-statistical directions of effect, these are not statistically significant. It could lead to nervousness about IMT for example given the unexpected and unexplainable direction of travel towards increasing mortality (whilst cutting LOS and complications)

We have removed all references to non-statistically significant results from the discussion.

5. I think my main concern is that we do not highlight the multimodal affect. I think this is probably the future (as it was in ERAS 20 years ago), this is the key concept going forward.

We have included a statement about the importance of multimodal prehabilitation in the Future research/clinical recommendations section.

Minor Concerns

1) 15 authors seems a lot to be listed for a meta-analysis – are they all justified? Could some be acknowledged rather than co-authors? (I am ok if not but it seems expansive)

This review would not have happened without a large team. In view of the above comment by the reviewer, we took the decision to acknowledge the contributions of all data extractors rather than include them as authors.

2) The word count seems quite high – could this be cut significantly (aprox 2500-3000 words is typical for a paper, though I accept that for a project as expansive as this it may be higher. If the paper word count could be cut it would improve the way it reads. Perhaps more in the supplementary section? We have reduced the word count to 4384 words.

3) First paragraph – saying up to 75% of patients undergoing major surgery is a negative view. I would say a typical rate (I tend to quote 40-50%) rather than the worst possible view. I accept why it was put but it increases fear about elective surgery which I am not sure is warranted. The statement is supported by evidence from the UK Postoperative Morbidity Survey and we prefer to leave it in.

4) Second paragraph – do we need to mention COVID? is this specifically relevant to the paper? We have removed references to COVID.

5) Third paragraph of introduction – please delete the bit – it's not needed. "e.g. exercise in cancer surgery 22; weight loss interventions in bariatric surgery 23; inspiratory muscle training in cardiac surgery 24; exercise in gastrointestinal cancer surgery 25 26, immunonutrition in head and neck surgery 27, etc."

We have deleted this section as the reviewer suggested.

6) Typically robotic radical prostatectomy is done one day, and the patient discharged the next, were these studies excluded?

We did not come across any studies in which patients underwent robotic radical prostatectomy. However, an inclusion criterion for our review was patients undergoing major elective surgery (under general anaesthesia resulting in a minimum hospital stay of at least 2 days) (see Methods section on page 7).

7) Methods are generally ok, but quite wordy. Similarly for the results. We have shortened both the methods and results sections.

8) Again COVID mentioned in discussion – is this appropriate? We have removed all references to COVID as the reviewer suggested.

9) The limitations section is very long – can you limit this? We have reduced this section substantially.

Reviewer: 2

Comments:

This is a large systematic review and meta-analysis including 144 randomized controlled trial to determine the risks and benefits of pre-admission interventions on postoperative outcomes in patients undergoing elective surgical procedures. The results demonstrate that some pre-admission interventions may reduce hospital stay and some postoperative complications.

First of all, I would like to appreciate the authors extensive work! Although of high interest, there are some major points which drew my attention.

- Trial selection includes trials from 1984 to 2019. The authors already state this as limitation of the trial. Why did the authors decide to include old trials in this analysis? Perioperative treatment has been changed in the last decades which impact patients' outcomes. I think this impedes the explanatory power of the results.

Our inclusion criteria were pre-specified (see Perry R, Scott LJ, Richards A, et al. Pre-admission interventions to improve outcome after elective surgery-protocol for a systematic review. Syst Rev 2016;5:88. doi: 10.1186/s13643-016-0266-9) and our review was intended to be comprehensive. However, we conducted sensitivity analyses for all interventions excluding trials published prior to 2010; these did not materially change the results.

- The analysis includes palliative surgical procedures. Please clarify this. Does this mean cancer surgery in general or really palliative surgical procedures. If palliative surgical procedures in a narrow sense, did the authors perform a subgroup analysis (palliative/non-palliative) to see whether this may distort the mortality results?

We did not specifically exclude palliative surgical procedures. However, very few studies included patients undergoing palliative surgery, so it would not have been possible to conduct a subgroup analysis. Our mortality analysis is 30 day mortality, which is more likely to be influenced by complications from surgery (which is what prehabilitation is trying to prevent) rather than the underlying disease.

- Most of the evidence grades demonstrate very low to low GRADE levels. Why did the authors include so many low evidence studies instead of focusing on larger trials with higher evidence levels? We have performed sensitivity analyses excluding trials at high risk of bias (see Online Supplementary Material).

- I would recommend to rewrite the results section. From my perspective, it is hard to read as in every paragraph details on where the information comes from are provided. I would recommend to more clearly present the results.

We have re-organised the results section to make it more readable.

- Please liaise with the policies of the journal, e.g., the discussion section is approximately 6 pages long.

We have shortened the discussion.

VERSION 2 – REVIEW

REVIEWER	Díaz-Alvarez, Agustín University Hospital of Salamanca
REVIEW RETURNED	18-May-2021

GENERAL COMMENTS	At the reviewer's request, I have stuck exclusively to the statistical aspect. After having read the article, the previous comments of other reviewers, as well as the modifications made by the authors, I believe that the manuscript is correct from a methodological point of view. Statistically, I believe it does not need any further correction.
---

REVIEWER	Pripp, Are Hugo
-----------------	-----------------

	Oslo universitetssykehus Ullevål, Oslo Centre for Biostatistics & Epidemiology
REVIEW RETURNED	21-May-2021

GENERAL COMMENTS	Statistical review; The study is relevant and comprehensive. I have mainly assessed the statistical analysis and it seems adequate, but have some few comments and concerns. Please specify the type of random (is it DerSimonian-Laird?) and fixed (is it inverse-variance?) meta-analysis model. I did not find any information about the type of software and/or meta-analysis packages used, please give detailed information. Publications bias could also be assessed with more formal statistical tests and methods as e.g. Egger-regression and trim-and-fill methods and heterogeneity could be assessed with a Cochrane Q test in addition to the I² statistics. Ideally, the computer codes to conduct the meta-analysis should be made available. It makes it much easier to check what type of meta-analysis methods you use and reproduce the analysis.
---

VERSION 2 – AUTHOR RESPONSE

Reviewer: 1

Dr. Agustín Díaz-Alvarez, University Hospital of Salamanca

Comments to the Author:

At the reviewer's request, I have stuck exclusively to the statistical aspect. After having read the article, the previous comments of other reviewers, as well as the modifications made by the authors, I believe that the manuscript is correct from a methodological point of view. Statistically, I believe it does not need any further correction.

Thank you for the positive comment.

Reviewer: 2

Dr. Are Hugo Pripp, Oslo universitetssykehus Ullevål

Comments to the Author:

Statistical review

The study is relevant and comprehensive. I have mainly assessed the statistical analysis and it seems adequate, but have some few comments and concerns.

Please specify the type of random (is it DerSimonian-Laird?) and fixed (is it inverse-variance?) meta-analysis model.

For dichotomous data we used the Mantel-Haenszel method to calculate the risk ratios and 95% CI for both random and fixed effects meta-analyses. For continuous data we used the inverse-variance method to calculate the pooled mean differences and 95% CI for both random and fixed effects meta-analyses. We have added these details in the data synthesis section on page 10.

I did not find any information about the type of software and/or meta-analysis packages used, please give detailed information.

All analyses were performed on Revman 5.3. We have added this information on page 11.

Publications bias could also be assessed with more formal statistical tests and methods as e.g. Egger-regression and trim-and-fill methods and heterogeneity could be assessed with a Cochrane Q test in addition to the I^2 statistics.

We have performed the Egger-regression tests as requested by the reviewer and inserted the results of these under Funnel plots (page 14).

The Cochran Q test is the χ^2 test shown in all our meta-analyses alongside the I^2 test.

Ideally, the computer codes to conduct the meta-analysis should be made available. It makes it much easier to check what type of meta-analysis methods you use and reproduce the analysis.

We agree with the reviewer that data from meta-analyses should be easily accessible by other researchers wishing to add to or replicate the results. However, both data entry and meta-analyses were conducted in RevMan and RevMan does not save the analysis code, making it difficult to replicate analyses in reviews. We are aware of a data tool (RevMan Parsing Tool for Reviewers, RAPTOR) which can extract data from RevMan files into an XML file which can then be easily imported into statistical packages such as R and Stata for repeat or further analyses. We would be happy to extract and share our data with other researchers upon request.